# Proposal for a physiotherapy assessment form for the evaluation of women patients with uro-gynecological disorders: A Delphi study

Ana González-Castro[1], Raquel Leirós-Rodríguez[2], Óscar Rodríguez-Nogueira[2], Mª José Álvarez-Álvarez[2], Arrate Pinto-Carral[2], Elena Andrade-Gómez[3] *

1 Nursing and Physical Therapy Department, University of León, Ponferrada, Spain, 2 SALBIS Research Group, Nursing and Physical Therapy Department, University of León, Ponferrada, Spain, 3 Nursing Department, University of La Rioja, La Rioja, Spain

☯ These authors contributed equally to this work.
* elena.andrade@unirioja.es

## Abstract

### Background

The correct selection of treatment techniques and methods in physiotherapy depends directly on a well-structured anamnesis, examination and assessment. Within urogynecological and obstetric physiotherapy there is no standardized and protocolized assessment that allows to follow established steps. For all this, the main objective of this study was to identify the assessment items that should be included in the a physiotherapeutic uro-gynecological assessment.

### Methods

Delphi study through a group of experts. Prior to this, a systematic search was carried out, accompanied by a review of grey literature, to obtain the possible items to be included in the forms. Subsequently, a Delphi study with two consecutive rounds of questionnaires was developed. A total of 6 expert physiotherapists participated in the study.

### Results

The initial questionnaire had 97 items and after two rounds one item was eliminated to obtain a total of 96 items in the final questionnaire.

### Conclusions

The experts agreed on most of the choices and finally obtained a standardized and protocolized assessment in uro-gynecological physiotherapy. Furthermore, this proposal should be considered by other professionals involved in the process of evaluation and treatment of pelvi-perineal alterations.

**Data Availability Statement:** The datasets used and analyzed during the current study are available in https://zenodo.org/records/10213486

**Funding:** This research was funded by the Professional Association of Physiotherapists of Castilla y León (Spain) (code: INV2022-30). The funders had no role in study design, data collection and analysis, decision to publish, or preparation of the manuscript.

**Competing interests:** The authors have declared that no competing interests exist.

## Introduction

Physiotherapy and its different areas of specialization have developed enormously in recent decades [1]. Based on the knowledge acquired in the Physiotherapy Degree, many physiotherapists decide to specialize in a health specialty such as urogynecology [1–6]. Uro-gynaecological physiotherapy can bring great benefits to women's health [7]. In fact, it has demonstrated a great capacity to improve the quality of life of patients with symptoms related to urinary and/or faecal incontinence, chronic pelvic pain and sexual disturbances [8–10]. In other words, it is a specialty that intervenes in functional alterations that cause serious consequences in the emotional health and social and personal relationships of these patients [7].

As in all health specialties, the correct choice of treatment and its management depend directly on a correct initial assessment [11]. However, in uro-gynecological physiotherapy, the assessment process is not protocolized, which leads to a greater inference of bias and a greater probability of errors on the part of physiotherapists. In addition, many women receive inappropriate treatment due to misdiagnosis [12].

In the anamnesis it is necessary to rigorously record the patient's daily activities, family and personal history, history of abdomino-pelvic surgery and relevant obstetric events in the patient's life [13–15]. For specific questioning in the pelvi-perineal field, it is important to reflect the reason for consultation and the aggravating and extenuating behaviors [16, 17]. When examining patients, it is necessary to organize a sequence of steps in order to ensure the most comfortable and least invasive examination for the patient, and the most efficient and simple for the professional [18]. On the other hand, the physical examination would not be complete without a correct postural and abdominal-pelvic assessment [19, 20]. Furthermore, in some cases it is necessary to obtain information through validated questionnaires as an important part of the initial assessment and re-evaluations [21–24].

It is now well known that standardization and optimization of clinical reasoning processes lead to increased efficiency in clinical practice [25]. Indeed, in 2020, the Chartered Society of Physiotherapy in London and the University of Southampton, also in the UK, defined research on "Diagnosis and prognosis: How is patient and/or physiotherapy treatment progress measured? And how is service performance measured and tested?" was among the top ten priorities for physiotherapy research [26]. Consequently, the development of a standardized assessment represents a strategy to facilitate clinical reasoning in physiotherapy.

To achieve this, the application of the Delphi method is a useful and effective strategy. This methodology has been used in physiotherapy in research on other pathologies that are also very prevalent, with positive consequences for the quality of life of patients and a high reduction of costs health services [27].

Among them, headaches stand out because of their high incidence and prevalence and because they are therefore considered a public health problem. This fact would therefore justify the importance of conducting research such as that carried out by Luedtke et al. [28]. In this study, a consensus was sought among experts in order to obtain the most useful screening tests for headache patients. At the methodological level, a preliminary systematic review was carried out to obtain the screening tests to be included in the Delphi study, followed by three consecutive rounds for the experts.

Taking all of the above into account, this research was developed in which the Delphi method was applied with the aim of identifying the assessment items that should be included in a physiotherapeutic uro-gynecological assessment. The ultimate aim was to create a new evaluation that would incorporate all existing tools while preserving the validity of existing assessment tools.

## Materials and methods

### Delphi study design and ethics statement

Prospective observational study based on the Delphi method with the aim of creating a standardized and protocolized uro-gynecological and obstetric assessment. The study was approved by the University of León Research Ethics Committee (code: 007–2022). All participants signed the informed consent for participation according to the Declaration of Helsinki (2018 version).

The Delphi method consists of the search for a common consensus among a group of experts on a specific topic [29, 30]. In this case, the Delphi study was conducted on the basis of a systematic review and then through two consecutive rounds of questionnaires.

The Delphi method justifies that the work of a group of experts yields better results than the work of individual experts [22–24]. For this reason, it is often very useful in the field of health, due to the difficulties often encountered in bringing together the professionals concerned in person. One of the objectives of this method, therefore, is to achieve a common consensus among experts on a specific topic [31].

### Participant and public involvement

Once the research had been designed, for the conduct of this research, all participants were informed of the aim of the research, its purpose and that the results obtained would be disseminated to the scientific community. This information was provided in a pre-participation briefing and was reflected in the informed consent to participate in the research.

In addition, participants received a report with the results obtained in this research and were invited to disseminate them.

### Previous systematic review

A systematic review was conducted through a literature search from 2016 to 2022 in the databases: PubMed, Scopus, ScienceDirect, Web of Science and Cinahl. The search terms used were Medical Subject Heading (MeSH) thesaurus keywords: *Physical therapy modalities*; *Rehabilitation*; *Diagnostic techniques and procedures*; *Obstetrical and gynecological diagnostic techniques*; *Urological diagnostic techniques*; *Gynecology*. This systematic review was used to obtain the terms that were subsequently included in the initial questionnaire of the Delphi study. In parallel, a grey literature search was carried out in the specialised publishers: Elsevier, McGraw Hill, Editorial Médica Panamericana, and Axon. The different literature search strategies used are described in Table 1.

### Expert group

A total of 41 physiotherapists were invited to participate in the study through a purposive and snowball sampling between 15 February and 31 March 2022. The inclusion criteria for participation in the study were: (a) being a graduate in physiotherapy; (b) belonging to a professional association of physiotherapists in Spain; (c) having complementary training in uro-gynecological physiotherapy (minimum duration of 75 hours); and (d) having a minimum of 4 years of clinical experience in the care of women patients with uro-gynecological alterations.

### Design of the initial questionnaire

The initial questionnaire began with a justification and description of the research, which participants agreed to take part in by signing the informed consent form. This was followed by a series of socio-demographic and professional questions about the participants that were used

**Table 1. Search equations used for the systematic review.**

| Database | Search equation |
|---|---|
| PubMed | ("Gynecology" [Mesh]) AND ("Diagnostic techniques and procedures" [Mesh]) AND ("Rehabilitation" [Mesh])<br>("Gynecology" [Mesh]) AND ("Diagnostic techniques and procedures" [Mesh]) AND ("Physical therapy modalities" [Mesh])<br>("Diagnostic techniques, obstetrical and gynecological" [Mesh]) AND ("Rehabilitation" [Mesh])<br>("Diagnostic techniques, obstetrical and gynecological" [Mesh]) AND ("Physical therapy modalities" [Mesh])<br>("Diagnostic techniques, urological" [Mesh]) AND ("Rehabilitation" [Mesh])<br>("Diagnostic techniques, urological" [Mesh]) AND ("Physical therapy modalities" [Mesh]) |
| Scopus | TITLE-ABS-KEY (Gynecology) AND TITLE-ABS-KEY (Diagnostic techniques and procedures) AND TITLE-ABS-KEY (Rehabilitation)<br>TITLE-ABS-KEY (Gynecology) AND TITLE-ABS-KEY (Diagnostic techniques and procedures) AND TITLE-ABS-KEY (Physical therapy modalities)<br>TITLE-ABS-KEY (Diagnostic techniques, obstetrical and gynecological) AND TITLE-ABS-KEY (Rehabilitation)<br>TITLE-ABS-KEY (Diagnostic techniques, obstetrical and gynecological) AND TITLE-ABS-KEY (Physical therapy modalities)<br>TITLE-ABS-KEY (Diagnostic techniques, urological) AND TITLE-ABS-KEY (Rehabilitation)<br>TITLE-ABS-KEY (Diagnostic techniques, urological) AND TITLE-ABS-KEY (Physical therapy modalities) |
| ScienceDirect | · (Gynecology) AND (Diagnostic techniques and procedures) AND (Rehabilitation)<br>(Gynecology) AND (Diagnostic techniques and procedures) AND (Physical therapy modalities)<br>(Diagnostic techniques, obstetrical and gynecological) AND (Rehabilitation)<br>(Diagnostic techniques, obstetrical and gynecological) AND (Physical therapy modalities)<br>(Diagnostic techniques, urological) AND (Rehabilitation)<br>(Diagnostic techniques, urological) AND (Physical therapy modalities) |
| Web of Science | TS = (Gynecology) AND TS = (Diagnostic techniques and procedures) AND TS = (Rehabilitation)<br>TS = (Gynecology) AND TS = (Diagnostic techniques and procedures) AND TS = (Physical therapy modalities)<br>TS = (Diagnostic techniques, obstetrical and gynecological) AND TS = (Rehabilitation)<br>TS = (Diagnostic techniques, obstetrical and gynecological) AND TS = (Physical therapy modalities)<br>TS = (Diagnostic techniques, urological) AND TS = (Rehabilitation)<br>TS = (Diagnostic techniques, urological) AND TS = (Physical therapy modalities) |
| CINAHL | (MH "Gynecology") AND (MH "Diagnostic techniques and procedures") AND (MH "Rehabilitation")<br>(MH "Gynecology") AND (MH "Diagnostic techniques and procedures") AND (MH "Physical therapy modalities")<br>(MH "Diagnostic techniques, obstetrical and gynecological") AND (MH "Rehabilitation")<br>(MH "Diagnostic techniques, obstetrical and gynecological") AND (MH "Physical therapy modalities")<br>(MH "Diagnostic techniques, urological") AND (MH "Rehabilitation")<br>(MH "Diagnostic techniques, urological") AND (MH "Physical therapy modalities") |

to characterize the sample (sex, age, year of qualification as a physiotherapist, length of professional experience in the uro-gynecological area and hours of specialized training in urogynecology).

At the end of the questionnaire, a final question was included asking for the participants' e-mail address to continue their participation in the second round of the study.

## Round I

The first-round questionnaire included all rating items obtained from the literature search previously conducted through the grey literature and systematic review.

In this first round, item scores were determined on a Likert scale from a score of 1 point (strongly disagree) to a maximum of 5 points (strongly agree).

Those items that scored 4 or 5 points in the first round in 80% or more of the cases were directly included in the final rating form. Items that scored 1 or 2 points in 80% or more of the cases were discarded directly from the form. Finally, items with a score of 3 were mostly selected for the second Delphi round.

### Round II

In the second-round questionnaire, items that did not obtain consensus among the participants in the first round were included in order to confirm their inclusion or exclusion from the final questionnaire.

In this second round, the items were evaluated through dichotomous questions about the need to include each item (Yes / No). In this case, items that achieved 80% or more positive scores were included in the assessment proposal resulting from this research.

### Statistical analysis

For the analysis of the continuous socio-demographic variables, a descriptive statistical analysis was carried out using the mean and standard deviation. For the analysis of qualitative variables and all rating items of the forms of both Delphi rounds a frequency analysis was performed.

All calculations were performed using the STATA software v.13 (Stata Corp., College Station, TX, USA).

## Results

### Results of the systematic review

A total of 873 results were identified from the literature search. Of these, 224 duplicates were eliminated and 127 were selected. Of these, 97 items were found to form the questionnaire for round I of the Delphi Study (Fig 1).

### Participation results

Of the 41 physiotherapists invited to participate in the study, eight participants completed round I of the Delphi study and of these eight, six completed round II.

The participants were predominantly female (92.3%) and their work experience ranged from two to 24 years (Table 2).

### Round I

Of the 97 items, 89 (91.8%) went directly into the final questionnaire, 8 (8.2%) were included in the round II questionnaire, and no items were eliminated. Among the items included for the second round were: seven screening items and one questionnaire and scales in uro-gynecological assessment.

### Round II

In this second round, one exploration item was eliminated with a score of more than 80% "No" from the participants. Therefore, from this second round, seven items went directly into the final questionnaire.

*Physiotherapeutic uro-gynecological assessment form proposal*

The proposed uro-gynecological assessment resulting from this research included a total of 96 items. Of these, 89 were obtained after round I and seven after round II (Fig 2).

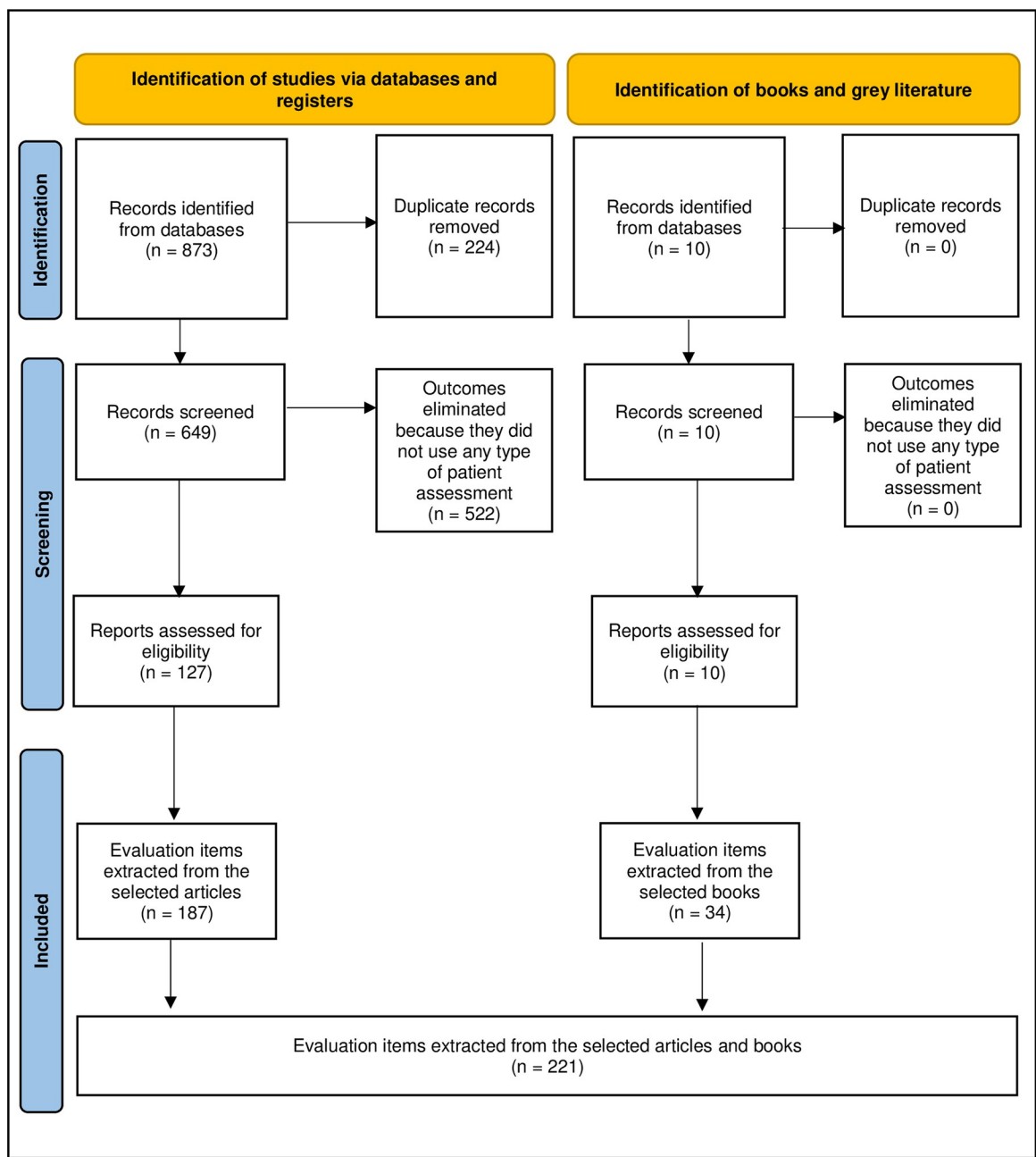

**Fig 1. Preferred Reporting Items for Systematic Reviews and Meta-Analyses (PRISMA) flow diagram.**

**Table 2. Characteristics of the participating experts.**

| Variable | mean ± standard deviation |
|---|---|
| Age (years) | 36.4 ± 7.8 |
| Seniority of qualification as physiotherapist (years) | 2008.2 ± 7.1 |
| Work experience (years) | 13.4 ± 7.3 |
| Specialised training in uro-gynecology (hours) | 334.4 ± 253.7 |

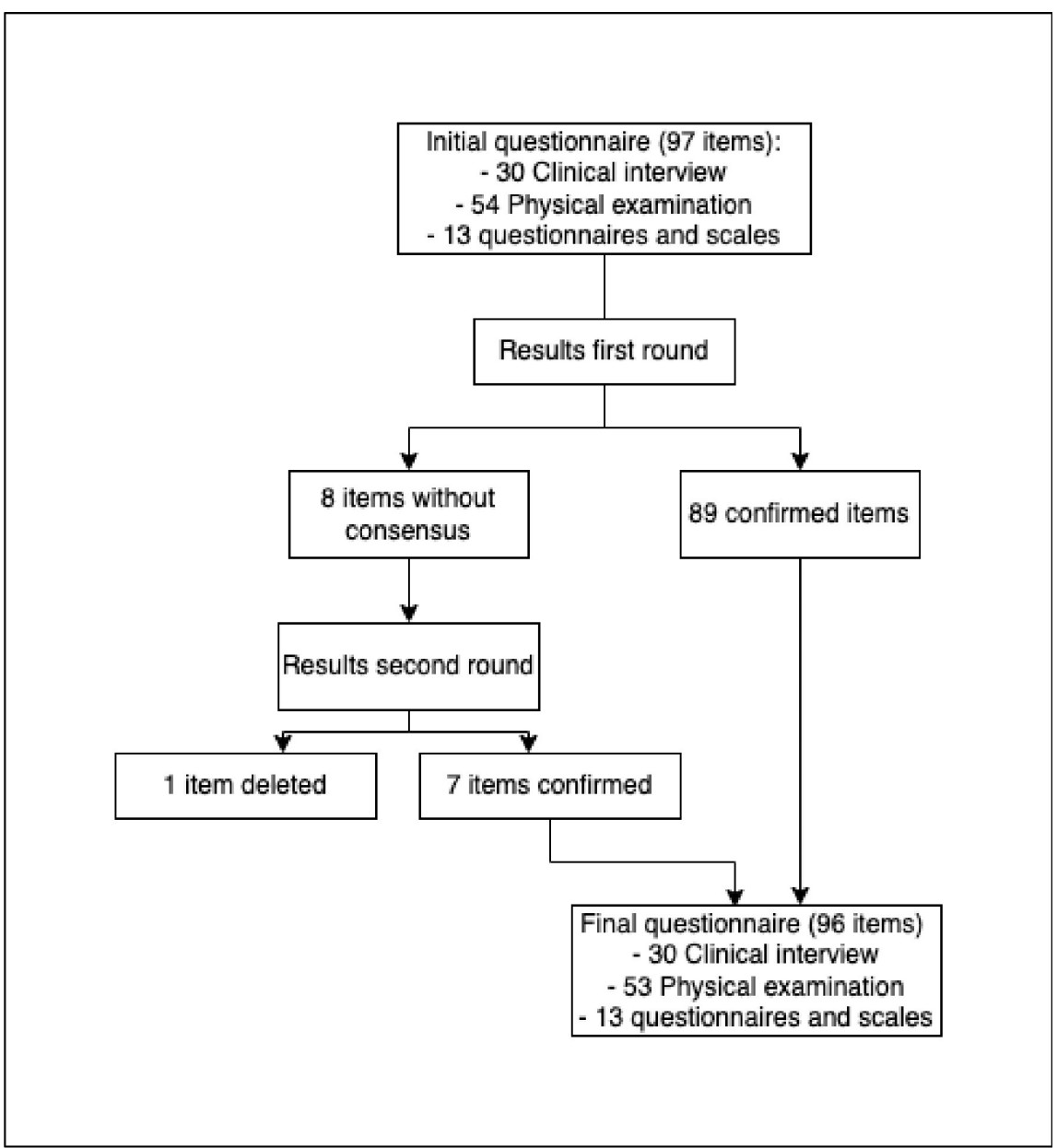

**Fig 2. Flow chart of item selection and deletion throughout the Delphi study.**

In terms of its components, the uro-gynecological assessment included 30 anamnesis items, 53 physical examination items and 13 items corresponding to specific questionnaires and scales (Table 3).

## Discussion

The aim of this study was to identify the assessment items that should be included in a uro-gynecological physiotherapy assessment. Following this study, a complete assessment was proposed, including the sections of the complete clinical interview, the abdomino-pelvic-perineal physical examination and the application of questionnaires and evaluation scales specific to this specialty.

**Table 3. Items included in the forms used in the two Delphi rounds and results obtained.**

| Item | First round results | | | Second round results | |
|---|---|---|---|---|---|
| | Including | Eliminated | Lack of consensus | Including | Eliminated |
| **Clinical interview** | | | | | |
| Personal details of the patient | Yes | | | | |
| Reason for consulting the physiotherapist | Yes | | | | |
| Number of previous pregnancies | Yes | | | | |
| Number and date of births | Yes | | | | |
| Family history | Yes | | | | |
| Personal medical history | Yes | | | | |
| Personal uro-gynecology history | Yes | | | | |
| Personal surgical history | Yes | | | | |
| Obstetric history | Yes | | | | |
| Date of onset of symptoms | Yes | | | | |
| Frequency of symptoms | Yes | | | | |
| Micturition rhythm/cycles | Yes | | | | |
| Characteristics of symptoms | Yes | | | | |
| Presence of incontinence and type of incontinence | Yes | | | | |
| Need for leakage protection and type | Yes | | | | |
| Hydration habits | Yes | | | | |
| Voiding habits | Yes | | | | |
| Presence of anal and/or faecal Incontinence and type | Yes | | | | |
| Frequency of leakage | Yes | | | | |
| Defecatory rhythm/cycles | Yes | | | | |
| Characteristics of leakage | Yes | | | | |
| Need for leakage protection and type | Yes | | | | |
| Feeding habits | Yes | | | | |
| Defecatory habits | Yes | | | | |
| **Exploration items** | | | | | |
| Flexion spine mobility test | Yes | | | | |
| Spinal column mobility test in extension | Yes | | | | |
| Spinal column mobility test in right and left lateral flexion | Yes | | | | |
| Spinal column mobility test in right and left rotation | Yes | | | | |
| Postural attitude in standing position | Yes | | | | |
| Postural attitude in seated position | Yes | | | | |
| Lasegue test | | | Dude | Yes | |
| Differential Lasegue test | | | Dude | Yes | |
| Bragard test | Yes | | | | |
| Iliac wing compression test | Yes | | | | |
| Rotes-Querolle test | | | Dude | Yes | |
| Gaenslen sign | | | Dude | Yes | |
| Mobility test of the sacro-iliac joints | Yes | | | | |
| Patrick test | | | Dude | Yes | |
| Guillet test | | | Dude | Yes | |
| G.Struyff quadrupedal test | | | Dude | | Deleted |
| Assessment of diaphragmatic tone | Yes | | | | |
| Assessment of diaphragmatic strength | Yes | | | | |
| Assessment of abdominal tone | Yes | | | | |

(*Continued*)

**Table 3.** (Continued)

| Item | First round results | | | Second round results | |
|---|---|---|---|---|---|
| | Including | Eliminated | Lack of consensus | Including | Eliminated |
| Assessment of abdominal strength | Yes | | | | |
| Abdominal eventration test | Yes | | | | |
| Sternal mobility test | Yes | | | | |
| Descending pressure test | Yes | | | | |
| Perineal descent test | Yes | | | | |
| Ano-coccygeal pressure test | Yes | | | | |
| Vulvar fork assessment | Yes | | | | |
| Vulvar and vaginal trophism assessment (staining) | Yes | | | | |
| Assessment of vaginal introitus and vaginal opening | Yes | | | | |
| Assessment of possible prolapse | Yes | | | | |
| Ano-vulvar assessment | Yes | | | | |
| Ano-pubic distance assessment | Yes | | | | |
| Assessment of possible scarring | Yes | | | | |
| Tone of the central nucleus of the perineum | Yes | | | | |
| Tone of the anal sphincter | Yes | | | | |
| Assessment of sensitivity | Yes | | | | |
| Assessment of bulbo-cavernosus reflex (S2-S4) | Yes | | | | |
| Cough reflex (D6-D12 / S3-S4) | Yes | | | | |
| Integration of the pelvic diaphragm into the body schema | Yes | | | | |
| Identification of descending perineum syndrome | Yes | | | | |
| Vaginal touch | Yes | | | | |
| Exploration of basal or global tone | Yes | | | | |
| Perineometry of ischio-cavernosus and bulbo-spongiosus muscles | Yes | | | | |
| Perineometry of the transverse muscle of the perineum | Yes | | | | |
| Perineometry of the pubo-vaginal muscle | Yes | | | | |
| Perineometry of the obturator internus muscle | Yes | | | | |
| Identification of parasitic muscle synergies | Yes | | | | |
| Presence of tone alterations | Yes | | | | |
| Presence of myofascial trigger points | Yes | | | | |
| Intravaginal scarring and possible fibrosis | Yes | | | | |
| Budin test | Yes | | | | |
| Bonney Manoeuvre | Yes | | | | |
| **Questionnaire and scales** | | | | | |
| International Consultation on Incontinence Questionnaire-Urinary Short Form | | | **Dude** | Yes | |
| Incontinence Severity Index | Yes | | | | |
| King's Health Questionnaire | Yes | | | | |
| Bladder Control Self-Assessment Questionnaire | Yes | | | | |
| Urogenital Distress Inventory-6 | Yes | | | | |
| Short version of the Incontinence Impact Questionnaire | Yes | | | | |
| Epidemiology of Prolapse and Incontinence Questionnaire | Yes | | | | |
| Pelvic Organ Prolapse, Incontinence and Sexual Questionnaire | Yes | | | | |
| Female Sexual Function Index | Yes | | | | |
| Impact of Female Chronic Pelvic Pain Questionnaire | Yes | | | | |
| Voiding calendar | Yes | | | | |
| Defecatory calendar | Yes | | | | |

The Delphi method has been applied in previous research in other physiotherapy specialties [32–35]. Among them, the research by Vitacca et al. [35] stands out because it is in line with the methodology applied in this research. In their case, their aim was to find a consensus among experts on pulmonary rehabilitation in patients with COVID-19 recovering from acute respiratory failure. In their research, on the other hand, they use a multidisciplinary group of experts composed of physiotherapists (specialized or not in pulmonology), pulmonologists, psychologists and methodologists. These authors, after the development of a previous systematic review and two Delphi rounds, obtained a number of items and proportion in their results very similar to those presented in the present study [35]. Furthermore, it is worth noting that the results obtained in both investigations obtained a high percentage of agreement among the participating experts.

At the same time, the congruence between the methodology followed in this research and that applied in previous research validates the results obtained [28, 36]. In the research conducted by Luedtke et al. [28] they also identified the low participation of experts. Considering that this was global research, only 17 experts out of the 20 invited participated in their research [28].

Despite the above examples, there are few publications focusing on uro-gynecological assessment and diagnosis that apply the Delphi method. In terms of their relationship, the publication by Dorey [37]. In it, he succeeded in generating a new classification system for male urinary incontinence. In his research, he used a group of 14 experts and was also able to generate new subjective and objective assessment questionnaires as well as various treatment options. The assessment questionnaire generated is notable for its similarity to the one proposed in this research: a set of anamnesis questions followed by screening items. Among the items included by Dorey are abdominal palpation, perineal exploration and assessment of pelvic floor muscle strength; also included in the uro-gynecological assessment proposal of the present research. Instruments of great importance for the evaluation of multidimensional constructs such as quality of life and female sexual function have been taken into account in this research. Specifically, through questionnaires such as the King's Health Questionnaire or the Incontinence Severity Index [38] for quality of life, and the Female Sexual Function Index [39] for female sexual function.

The authors must acknowledge that the low participation rate in the study limits the extrapolation of results and exposes poor adherence by participants. The factors influencing adherence to the questionnaires are not yet fully defined. It has been proposed that the visual aspects of a questionnaire do not affect the rate and quality of responses [40]. At the same time, a recommendation to increase the participation rate in online questionnaires has also been defined as a recommendation to make personalized invitations and souvenirs for each participant on a weekly basis [41]. However, despite the low participation rate in the research, the participants' responses provided a high rate of consensus regarding the items to be included and removed from the questionnaire. Finally, we must also recognize as a limitation that this Delphi study had as a sample only physiotherapy professionals who are experts in this specialty and that, above all, for future validations of this assessment proposal, the opinion of the patients could be taken into account.

At the same time, this research also has important strengths. All participants in the study met the stringent inclusion criteria defined, including long work experience and specialized training in the field. These characteristics increase the reliability of the results obtained in this research. Finally, the fact that the questionnaire was created online can be considered a strength because of the ease with which it can be modified and improved in the future. Now, after the design of this tool, it should be implemented in the clinical setting to confirm its reliability and usability by professionals. As well as the patients' perception of its content.

## Conclusions

This Delphi study demonstrated that a standardized and protocolized uro-gynecological physiotherapy assessment can be created to facilitate clinical reasoning in this area of specialization. The proposed assessment includes 96 items: 30 anamnesis items, 53 examination items and 13 items corresponding to questionnaires and scales. Consequently, the variables necessary to carry out a complete and objective assessment method in the specialty of urogynecological physiotherapy have been identified. This proposal should be taken into account by professionals in this specialty involved in the assessment and treatment of patients with pelvi-perineal alterations.

## Author Contributions

**Conceptualization:** Ana González-Castro, Raquel Leirós-Rodríguez, Óscar Rodríguez-Nogueira, Mª José Álvarez-Álvarez, Arrate Pinto-Carral, Elena Andrade-Gómez.

**Data curation:** Ana González-Castro, Raquel Leirós-Rodríguez, Óscar Rodríguez-Nogueira, Mª José Álvarez-Álvarez, Arrate Pinto-Carral, Elena Andrade-Gómez.

**Formal analysis:** Ana González-Castro, Raquel Leirós-Rodríguez, Óscar Rodríguez-Nogueira, Mª José Álvarez-Álvarez, Arrate Pinto-Carral, Elena Andrade-Gómez.

**Funding acquisition:** Elena Andrade-Gómez.

**Investigation:** Ana González-Castro, Raquel Leirós-Rodríguez, Óscar Rodríguez-Nogueira, Mª José Álvarez-Álvarez, Arrate Pinto-Carral, Elena Andrade-Gómez.

**Methodology:** Ana González-Castro, Raquel Leirós-Rodríguez, Óscar Rodríguez-Nogueira, Mª José Álvarez-Álvarez, Arrate Pinto-Carral, Elena Andrade-Gómez.

**Resources:** Ana González-Castro, Raquel Leirós-Rodríguez, Óscar Rodríguez-Nogueira, Mª José Álvarez-Álvarez, Arrate Pinto-Carral, Elena Andrade-Gómez.

**Software:** Ana González-Castro, Raquel Leirós-Rodríguez, Óscar Rodríguez-Nogueira, Mª José Álvarez-Álvarez, Arrate Pinto-Carral, Elena Andrade-Gómez.

**Supervision:** Ana González-Castro, Raquel Leirós-Rodríguez, Óscar Rodríguez-Nogueira, Mª José Álvarez-Álvarez, Arrate Pinto-Carral, Elena Andrade-Gómez.

**Validation:** Ana González-Castro, Raquel Leirós-Rodríguez, Óscar Rodríguez-Nogueira, Mª José Álvarez-Álvarez, Arrate Pinto-Carral, Elena Andrade-Gómez.

**Visualization:** Ana González-Castro, Raquel Leirós-Rodríguez, Óscar Rodríguez-Nogueira, Mª José Álvarez-Álvarez, Arrate Pinto-Carral, Elena Andrade-Gómez.

**Writing – original draft:** Ana González-Castro, Raquel Leirós-Rodríguez, Óscar Rodríguez-Nogueira, Mª José Álvarez-Álvarez, Arrate Pinto-Carral, Elena Andrade-Gómez.

**Writing – review & editing:** Ana González-Castro, Raquel Leirós-Rodríguez, Óscar Rodríguez-Nogueira, Mª José Álvarez-Álvarez, Arrate Pinto-Carral, Elena Andrade-Gómez.

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
