## [Decision Letter · Decision Letter 0]

2 Nov 2023

PONE-D-23-26905Proposal for a physiotherapy assessment form for the evaluation of women patients with uro-gynecological disorders: a Delphi studyPLOS ONE

Dear Dr. Leirós-Rodríguez,

Thank you for submitting your manuscript to PLOS ONE. After careful consideration, we feel that it has merit but does not fully meet PLOS ONE’s publication criteria as it currently stands. Therefore, we invite you to submit a revised version of the manuscript that addresses the points raised during the review process.

We look forward to receiving your revised manuscript.

Kind regards,

Malgorzata Wojcik, Ph.D

Academic Editor

PLOS ONE

Journal Requirements:

"This research was funded by the Professional Association of Physiotherapists of Castilla y León (Spain) (code: INV2022-30). "

Additional Editor Comments:

Dear Authors,

The article is interesting however, it requires corrections. Please make them in accordance with the reviewer's comments.

best wishes

Małgorzata Wójcik

Reviewers' comments:

Reviewer's Responses to Questions

**Comments to the Author**

1. Is the manuscript technically sound, and do the data support the conclusions?

Reviewer #1: Partly

2. Has the statistical analysis been performed appropriately and rigorously? 

Reviewer #1: I Don't Know

3. Have the authors made all data underlying the findings in their manuscript fully available?

Reviewer #1: Yes

4. Is the manuscript presented in an intelligible fashion and written in standard English?

Reviewer #1: Yes

5. Review Comments to the Author

Reviewer #1: Thank you for the opportunity to review this paper. It is an interesting topic and of value to those working in this area and to patients experiencing uro-gynaecological disorders.

The main issue with the paper is the lack of context. Although the authors write extensively about the importance of the issue, they have not stated whether this is a problem identified by patients with uro-gynaecological disorders or by physiotherapists. I cannot see any reference to patient engagement in the design, conduct, or dissemination of the study so assume it is a topic identified by physiotherapists or academics. I wonder if patient views were sought and informed the choice of topic as this would seem to be a powerful voice given the reported experience and poor health outcomes for those with such disorders. The authors allude to improving the patient experience, but it does not seem to be an explicit aim of their study. Page 10 states ‘the methodology has been used ------ with serious consequences for the quality of life for patients and a high economic impact on health services’ but it is not clear if this refers to a positive or negative impact and how they intended the current study to impact on patients.

In the section on participant and public involvement, reference is made to stakeholder i.e. physiotherapist involvement but it is not clear if this was as part of the design or conduct of the study. It was also unclear whether the purpose of the study was to rationalise the range of existing assessment tools for this disorder or to create one new assessment that incorporated all the existing tools. Has the new assessment retained the validity of the existing assessment tools, which presumably have been validated and tested, or have amendments been made to some items. Any implications relating to permissions or intellectual property by the creation of the new proposed assessment need to be noted.

The results of the systematic review indicate there were 873 papers retrieved of which 224 were duplicates and were removed. 127 papers were included in the review. There would appear to be another 653 papers retrieved but there is no indication of how they were dealt with. It would be useful to include a PRISMA statement and diagram mapping the stages of the systematic review. It was also unclear whether any papers from the grey literature were included in the review and how they were quality assessed.

The response rate of 8 out of 41 (reducing to 6/41) in terms of participation is low and it would be helpful to include observations about why this occurred. Despite the justification and references to other similar studies cited in the discussion, it is questionable that there can be confidence in the results of the Delphi study given the low response. The authors refer to a previous study where the response rate had been low but in fact that study included 17 out of a possible sample of 20 which is a considerably higher percentage than the current study.

One of the major concerns about the low engagement of the stakeholders is their interest in developing the tool vis a vis my earlier comments about the context for the study and its possible implementation. Discussion with the stakeholders, including patients, would have increased the potential for the research to be perceived as relevant to them and would potentially have improved recruitment.

A 96 item questionnaire seems very extensive and potentially cumbersome to complete in a consultation and clinical environment. There is no indication if the complete assessment tool has been piloted or whether participants views have been sought on the size and practicality of delivering an assessment. It would be helpful to have some context of how it is intended to be administered. P17 refers to possible poor adherence but does not explore this in any depth. There is no reference to the next steps for the assessment tool, plans for implementation, the likely impact of the tool, the patient perspective of such an extensive assessment.

I hope the authors find the comments useful in revising their paper.

6. PLOS authors have the option to publish the peer review history of their article (what does this mean?). If published, this will include your full peer review and any attached files.

Reviewer #1: **Yes: **Dr Virginia Minogue

---

## [Author Response · Author response to Decision Letter 0]

30 Nov 2023

Dear Reviewers and Editor of PLOS One:

Thank you very much for your suggestions and contributions to improve the quality of the manuscript. Following your indications, we respond, point by point, to the reviewers' comments.

In the text, all the modified or added sentences have been written in red to facilitate the correction by the reviewers.

Editor comments:

1. Thank you for providing your data set using a repository. If you could please translate this data into English as per PLOS Data Regulations we ask for data to be ‘fully available without restriction’.

The Authors have published in Zenodo the data related to the two rounds of the Delphi study carried out through two Excel files fully translated into English.

Once again, thank you very much for the time spent and the interest shown in this work; as well as in the positive evaluations you have given of it.

Receive a warm greeting,

The authors.

---

## [Decision Letter · Decision Letter 1]

6 Dec 2023

Proposal for a physiotherapy assessment form for the evaluation of women patients with uro-gynecological disorders: a Delphi study

PONE-D-23-26905R1

Dear Dr.Raquel Leirós-Rodríguez,

We’re pleased to inform you that your manuscript has been judged scientifically suitable for publication and will be formally accepted for publication once it meets all outstanding technical requirements.

Kind regards,

Malgorzata Wojcik, Ph.D

Academic Editor

PLOS ONE

Additional Editor Comments (optional):

Reviewers' comments:

Reviewer's Responses to Questions

**Comments to the Author**

1. If the authors have adequately addressed your comments raised in a previous round of review and you feel that this manuscript is now acceptable for publication, you may indicate that here to bypass the “Comments to the Author” section, enter your conflict of interest statement in the “Confidential to Editor” section, and submit your "Accept" recommendation.

Reviewer #2: (No Response)

2. Is the manuscript technically sound, and do the data support the conclusions?

Reviewer #2: Yes

3. Has the statistical analysis been performed appropriately and rigorously? 

Reviewer #2: Yes

4. Have the authors made all data underlying the findings in their manuscript fully available?

Reviewer #2: Yes

5. Is the manuscript presented in an intelligible fashion and written in standard English?

Reviewer #2: Yes

6. Review Comments to the Author

Reviewer #2: The article draws attention to the need to standardize the interview and examination during urogynecological physiotherapy. A detailed analysis of many works allowed us to isolate the most important elements necessary to prepare a research questionnaire.

7. PLOS authors have the option to publish the peer review history of their article (what does this mean?). If published, this will include your full peer review and any attached files.

Reviewer #2: **Yes: **Magdalena Dębińska

---

## [Editor Report · Acceptance letter]

15 Dec 2023

PONE-D-23-26905R1 

PLOS ONE

Dear Dr. Leirós-Rodríguez, 

I'm pleased to inform you that your manuscript has been deemed suitable for publication in PLOS ONE. Congratulations! Your manuscript is now being handed over to our production team.

Kind regards, 

on behalf of

Dr. Malgorzata Wojcik 

Academic Editor

PLOS ONE